# Taking the Long View for Oceans and Human Health Connection through Community Driven Science

**DOI:** 10.3390/ijerph18052662

**Published:** 2021-03-06

**Authors:** Usha Varanasi, Vera L. Trainer, Ervin Joe Schumacker

**Affiliations:** 1School of Aquatic and Fishery Sciences, College of the Environment, University of Washington, Seattle, WA 98195, USA; 2Environmental and Fisheries Science Division, National Marine Fisheries Service, Northwest Fisheries Science Center, National Oceanic and Atmospheric Administration, 2725 Montlake Blvd. E., Seattle, WA 98112, USA; vera.l.trainer@noaa.gov; 3Department of Fisheries, Quinault Indian Nation, Taholah, WA 98587, USA; jschumacker@quinault.org

**Keywords:** oceans, human health, indigenous, nature, harmful algal blooms, reciprocal healing

## Abstract

The most proactive approach to resolving current health and climate crises will require a long view, focused on establishing and fostering partnerships to identify and eliminate root causes of the disconnect between humans and nature. We describe the lessons learned through a unique scientific partnership that addresses a specific crisis, harmful algal blooms (HABs), along the northeast Pacific Ocean coast, that blends current-day technology with observational knowledge of Indigenous communities. This integrative scientific strategy resulted in creative solutions for forecasting and managing HAB risk in the Pacific Northwest as a part of the US Ocean and Human Health (OHH) program. Specific OHH projects focused on: (1) understanding genetic responses of tribal members to toxins in the marine environment, (2) knowledge sharing by elders during youth camps; (3) establishing an early warning program to alert resource managers of HABs are explicit examples of proactive strategies used to address environmental problems. The research and monitoring projects with tribal communities taught the collaborating non-Indigenous scientists the value of reciprocity, highlighting both the benefits from and protection of oceans that promote our well-being. Effective global oceans and human health initiatives require a collective action that gives equal respect to all voices to promote forward thinking solutions for ocean health.

## 1. Introduction

The confluence of a global pandemic and intensifying climate change has increased interest and yearning for contact with nature (wilderness, green and blue spaces). The growing awareness about how important nature is to human well-being could be a game-changer in public attitudes toward the natural world [1,2,3,4]. Could this be a powerful impetus for shifting our focus from a short-term approach to taking a long, proactive view? A perfect scenario would include a long-view strategy to identify and eliminate root causes of the disturbance of our relationship with nature while still allowing us to respond to current crises with short-term technological approaches. In this way, industrialized society would implement proactive strategies, beyond our customary methods of addressing environmental and human health problems ‘after the fact’. We must work to broaden our current oceans and environmental laws and mandates that focus largely on depleted species and degraded states to include all vulnerable ecosystems. Over the last two decades, new initiatives (e.g., One Health, Nature and Health, Oceans and Human Health) and many scholarly articles and studies have highlighted the benefits of nature contact to human health, but these studies are primarily investigating how nature exposure benefits human health and well-being (reviewed by [5,6]). There is a growing movement calling for *reciprocal* healing of nature and people [1,2,3,4,7,8]. However, it is not within the scope of this paper to exhaustively review this literature. Based on our experience as described in Varanasi [3], we caution against an overly human-centric view of how nature heals us (nature *for* health paradigm) and emphasize the importance of shifting individual and societal attention to preventive measures to maintain both human and ecological health (nature *and* health paradigm). Our long history of dealing with environmental crises and degradation of aquatic ecosystems as well as studies of terrestrial ecosystems has convinced us that once an aquatic or terrestrial ecosystem is damaged, it is nearly impossible to return it to its original state [9,10,11,12].

In this paper, we focus on aquatic ecosystems, accepting humans as a part of this natural system, to explore the critical question: how can current-day technological advances be combined with historical knowledge, exemplified by traditional, Indigenous wisdom based on observations of natural systems, to result in a rich relationship of learning, discovery, and creative solutions for oceans and human health?

As oceans comprise almost three fourths of the Earth’s surface, humans depend on the world’s seas in more ways than any other ecosystem. Rebalancing our relationship with oceans can be informed by learning from and supporting Indigenous peoples’ connection with the sea. Indigenous knowledge about conservation and resource management is being recognized with growing respect [1,13,14,15], especially realizing that Indigenous people successfully manage over 40% of Earth’s ecosystems [16,17]. Biodiversity is among the highest on lands managed by Indigenous communities, indicating the importance of these lands in survival of species [17].

Ocean scientists, lawmakers, industries committed to reducing and resolving the destruction faced by our world’s ocean(s), managers of natural resources, and the public are coming together during the 2021–2030 United Nations (UN) Decade of Ocean Science for Sustainable Development (https://www.oceandecade.org/; accessed on 4 March 2021) to design ocean solutions to combat climate change and environmental degradation. This initiative stems from the conviction that oceans will benefit from people who are inspired to implement conservation practices and advocate for positive change. Indeed, oceans and humans are inextricably linked: Our activities on land, sea, and in the air affect ocean health and, in turn, healthy oceans positively affect human health and well-being [7,18]. The belief that, because of its vastness and depth, oceans can tolerate any insult is clearly erroneous as serious health threats from harmful bacteria, algal bloom toxins, chemical pollutants, and diseases can be transferred from aquatic wildlife to people [19]. We believe that bringing together experts of all backgrounds, Indigenous and non-Indigenous communities alike, to implement co-created oceans coalitions, is an important part of a solution that studies people and nature co-existing harmoniously. Optimal, lasting protections of coastal ocean areas (i.e., marine protected areas) are accomplished by combining top-down regulation with bottom-up stewardship from local and Indigenous communities [20]. It is very difficult to enforce restrictions on activities such as shipping, fishing, dumping without local eyes on the ocean(s). Place-based coastal communities are true stewards of the ocean(s).

Industrialized societies have become accustomed to seeking quick fixes and technological solutions after a human health crisis or environmental health disaster. The most typical approach remains largely focused on post-traumatic solutions such as cleaning up polluted waters, controlling algal blooms, or even developing vaccines or treatments after a pandemic, rather than taking a longer, more proactive, and holistic approach toward environmental conservation and disease avoidance. We propose one solution to this siloed approach, calling for the inclusion of the concept of biocultural stewardship [21]. This biocultural approach emphasizes the active co-creation and mutual interdependence of biological and cultural diversity [22,23] into complementary, relational values of humans with one another and with nature [24]. This construct recognizes the intersection of the health of natural resources, e.g., oceans and coastal environments, with human health and well-being. Such a philosophy has been an integral part of the lives of place-based, Indigenous coastal people for centuries [25,26], where a “more-than-human-world” is a key concept [27]. For example, in contrast to the non-Indigenous view of water as a resource, something that can be used or exploited, many Indigenous communities view water (and oceans) as a living entity to which they have a sacred responsibility (e.g., [28]). The Indigenous approach that highlights a balanced strategy to ocean resource harvest and protection was not considered in past ocean management plans. However, responsible management was recognized by the landmark “Boldt Decision” court case (*United States v. Washington*, 384 F. Supp. 312 (W.D. Wash. 1974), aff’d, 520 F.2d 676 (9th Cir. 1975)), which reaffirmed the treaty-fishing rights of Washington State tribes while also recognizing individual tribes’ sacred histories of management and conservation. The Quinault and the Yakama Tribes were recognized as “self-regulatory” because of their demonstrated history of self-management and conservation of salmon stocks. A biocultural framework helps us “relearn Indigenous understandings of alternative ways for humans to relate with the natural world” (e.g., [21]). Thus, our current day ‘crisis culture’ can be changed by engaging biocultural approaches and supporting tribal and Indigenous sovereignty through co-management of ocean resources.

To expound on this thesis, we describe a partnership with coastal tribes of Washington State, as a part of the US Ocean and Human Health (OHH) program in the Pacific Northwest (2004–2009), which included an emphasis on collaborative harmful algal bloom (HAB) research and monitoring. This HAB project built upon the initial National Oceanic and Atmospheric Administration (NOAA) National Centers for Coastal Ocean Science Monitoring and Event Response to HABs (NCCOS MERHAB) funding that was requested first by the Quileute Tribe as a need for establishing a system of solutions to cope with increasing HAB problems on the Washington coast. This alliance resulted in a joint proposal (2000–2005) among several science organizations [29] to develop an optimized HAB monitoring partnership, that resulted in the Olympic Region Harmful Algal Bloom (ORHAB) project. These HABs provide a vital example of the interdependence of humans and nature. Some blooms can be exacerbated by anthropogenic nutrient inputs (*humans impacting nature*), while shellfish toxicity due to HABs can cause human illness (*nature impacting humans*). Coast Salish tribes from the Pacific Northwest and other coastal, native communities have centuries of experience with HABs. One of the first written records of a poisoning event occurred in 1799 when over 100 Aleut tribal members died after eating toxic blue mussels while on a Russian fur trading expedition [30]. Our OHH partnership began at a time when HAB outbreaks on the west coast were growing significantly, creating an opportunity for active, real-time sharing of traditional ecological knowledge by our experienced Coast Salish partners. Demonstrating the unique and complementary contributions of all researchers, a joint funding proposal was submitted at the start of the ORHAB project, laying the groundwork for a collaboration rooted in mutual respect and regard that continues to thrive today. Tribal wisdom was shared, not as an exotic relic of the past, but as an illustration of what is possible if we learn to focus our attention on the interdependence and reciprocity of Nature with humans, a lens that causes us to take the longer, more holistic view.

## 2. History of US Ocean and Human Health Program in the Pacific Northwest

The West Coast Center for Oceans and Human Health (West Coast Center) was one of three national Centers of Excellence established by NOAA in 2004 [31]. The West Coast Center was housed at the Northwest Fisheries Science Center which built upon its existing expertise in the study of pathogens and toxin-producing algae that affect shellfish, fish, and marine mammals under varying environmental conditions, including climate change. The overall goal of the West Coast Center was to understand, predict and reduce direct and indirect risks to human health caused by HAB contamination of coastal waters, and to assist natural resource and human health managers to deal with these challenges. Specific research responsibilities included developing early warning systems, optimizing benefits and minimizing risks from the sea, and fostering the OHH community.

### 2.1. OHH Communities in the Pacific Northwest

The partnership of NOAA scientists with coastal tribes generated several lessons for present and future scientific endeavors. Here we describe three such joint studies and the lessons learned, demonstrating proactive strategy toward addressing environmental problems. Specific OHH projects focused on understanding genetic responses tribal members to toxins in the marine environment, knowledge sharing by elders during youth camps, and establishing an early warning program to alert resource managers of HABs.

Coastal tribes are the cornerstone collaborators of the OHH community in the Pacific Northwest (PNW). For centuries, native people in this region have lived interdependently with nature. Tribal participants in OHH invited us to learn more about the local ocean ecosystem through projects and partnerships that shared common goals with us. Members of the Makah Tribe spoke about a highly productive ocean region in their native seas, near the border of the U.S. and Canada, which they call The Prairie. The ORHAB and OHH partnerships confirmed that The Prairie is a highly productive site with abundant phytoplankton food available for higher levels of the marine food chain. However, among the phytoplankton species were some that can form HABs. Humans can be harmed when these HABs are transported by currents to the Washington coast where they are concentrated in shellfish, such as razor clams and Dungeness crabs, which are highly desirable food items, and necessary subsistence foods for coastal tribes who often live in remote regions. Although NOAA scientists believed that the origination of phytoplankton blooms at this site was a new discovery, the Makah Tribe explained what they had known for centuries: this is a site where the sea is bountiful; fishing for salmon and halibut is fruitful; marine birds are plentiful; and whales are commonly seen. That this site was an origination site or a “birthplace” of HABs was important information that eventually led to enhanced protection of seafood safety in coastal Washington State. This critical knowledge added to the overall strategy to promote seafood safety for coastal communities, including tribes, by providing an early warning system for blooms developing in The Prairie.

Another collaborative discovery of the OHH partnership focused on how organisms, including humans, adapt to toxins in their environment. The Makah and other coastal tribes have a saying, “When the tide is out, the table is set”. In Neah Bay, Washington, where many Makah tribal members reside, shellfish, a major source of protein, are a subsistence food. Nearly 70% of households participate in shellfish harvest activities [32]. Members of the Makah Tribe tell the story of potlatches (tribal ceremonial feasts) where visitors would come from over the mountains to share in the seafood bounty [32,33]. These stories suggest that the visitors became ill from consuming shellfish which contained toxins, while the Makah, who habitually rely on shellfish in their diet, did not. One anthropological study reported that members of the Makah Tribe have adapted to shellfish poisoning due to their long history of exposure to shellfish in their diet [32]. These traditional stories provided inspiration to further our understanding of genetic resistance to toxins in shellfish that consume toxic algae [34], leading to a collaboration with the Makah Tribe to understand whether people, such as the Makah, who consumed shellfish also demonstrated resistance to toxins. This study showed that, in contrast to shellfish which can feed on and accumulate toxic algae and survive [32,34], tribal members do not appear to be resistant to toxins in their shellfish food [33]. This knowledge informed an optimal strategy for human health protection and outreach associated with HABs and their toxins.

### 2.2. Partnering with Coastal Tribes

#### 2.2.1. Partnering to Share Knowledge

Our partnership with Indigenous scientists and communities has provided valuable lessons that can guide future collaborations among organizations across the globe (Box 1). As part of the effort to foster community interactions with PNW coastal tribes whose diet, health and livelihood are strongly dependent on the coastal ocean, NOAA scientists teamed with tribal and state scientists and cultural experts of the Quinault Indian Nation and the Makah Tribe to offer summer camps for middle school students (*Lesson 1, Share Knowledge*). The overall goals of the camp were to improve ocean literacy, create enthusiasm and opportunity for hands on marine sciences, and encourage student empowerment. Part of the curriculum included listening to tribal elders tell traditional stories of the sea, including the observation of bioluminescent waters that warned of toxic algae. These listening events helped the tribal students see the wealth of traditional knowledge among the elders. It further demonstrated that the telling of the stories was a transfer of scientific and observational knowledge, proving its firm position and importance at the center of the non-Indigenous scientific canon.

Box 1Lessons to guide the long view approach for oceans and human health.1. Share knowledge2. Listen3. Foster collaborative and equitable relationships from the inception of an idea4. Conduct science as a collective action5. Cultivate mutual respect

#### 2.2.2. Partnering to Conduct Scientific Studies

The West Coast Center approached coastal tribes with a proposal for a study of HABs, after learning of the need for a collaborative monitoring program. When speaking of this project to tribal elders and shellfish managers, they listened without commenting (*Lesson 2, Listen*). They continued to listen but offered their excitement only when we moved from telling them what we knew to talking about sharing knowledge. We suggested using new scientific tools and equipment together with all coastal tribes, including identification of phytoplankton using microscopy. This started the development of joint research proposals, first to MERHAB, then to OHH, which encompassed the viewpoints and scientific needs of all partners. Working collaboratively to learn new methods, such as testing for toxins in seafood and seawater using specific immunological assays, allowed the participant tribes to make decisions about the safety of their own harvest by conducting assays in their own, newly established labs. Currently, the Quinault Indian Nation, the Quileute Tribe, and Makah Tribe all have independent HAB toxin testing labs that were established by sharing ideas used to write successful grant proposals. Through this collaboration, we developed the Olympic Region Harmful Algal Bloom (ORHAB) Partnership (Figure 1) in response to sudden closures of shellfish harvest by the Washington State Department of Health due to poorly understood outbreaks of marine biotoxins including paralytic shellfish toxins and domoic acid (*Lesson 3, Foster Collaborative and Equitable Relationships from the Inception of an Idea*). State resource managers joined with ocean scientists, including NOAA, the University of Washington Olympic Natural Resources Center, and members of the Quileute Tribe, the Quinault Indian Nation, and the Makah Tribe (Figure 1), to protect public health on the Washington coast by establishing a comprehensive monitoring and research program to better understand the underlying dynamics of these toxin outbreaks. This ORHAB partnership led to a deeper understanding of the environmental drivers of HABs and the early warning indicators of shellfish toxicity (Figure 1).

This type of understanding is critical to taking a long view in resource management of wilderness, including blue and green spaces. When natural resource agencies and policy makers shift their focus from reactive responses to the long view of environmental and human health crisis toward better management, it is hoped that sufficient attention and assets will be devoted for the timely development of tests and indicators. These provide early warning of an impending environmental crisis and understanding of the causes to help reduce or eliminate risks to promote healthy ecosystems of which humans are an integral part [3,11,12]. In the specific case of HABs, it is critical to identify the sites that serve as the source of these blooms while learning how environmental factors (e.g., wind, temperature, nutrient loading, and pH) affect the toxicity. When we use a broader lens devoted to look for the root causes of environmental drivers of ocean challenges, a paradigm shift from crisis management to a longer, better informed, ‘up-stream’ response is achieved. In the end, this approach will save resources, and protect the overall health of humans and oceans.

To develop funding and ownership at the State level, these research efforts were transitioned from NOAA to State funding in 2003 through a surcharge to Washington State shellfish licenses (RCW 77.32.555), allowing the ORHAB partnership and its associated monitoring to continue well into the future. This is a clear example of inclusive science, not extractive, “helicopter” science, that ultimately was used to inform legislators to enact change that benefited these coastal communities, promoting environmental justice [35]. Through mutual learning facilitated by the cornerstone ORHAB project, the Makah Tribe, the Quinault Indian Nation, and the Quileute Tribe have developed and now support their own internal monitoring programs that work cooperatively with state and academic partners (*Lesson 4, Conduct Science as Collective Action*). This PNW HAB Bulletin (Figure 2), an early warning system that is built upon the foundation of the ORHAB partnership, is in the process of becoming operational as the foremost advanced warning system for HABs in the northeast Pacific Ocean [36].

#### 2.2.3. Partnering to Make Resource Management Decisions

In October 1991, the first-ever domoic acid closure of razor clams and Dungeness crab harvest on the Washington coast [37] closed beaches to recreational and commercial shellfish harvesting [38], an economic and cultural blow to local fishing communities including coastal Tribal communities. Domoic acid, the toxin produced by the marine phytoplankton genus, *Pseudo-nitzschia*, impacts the health of both humans and marine wildlife, including whales, sea lions, sea otters and marine birds, by permanently damaging the nerve receptors responsible for memory and learning. Humans, marine mammals and marine birds share the same symptoms during these poisoning events, including gastrointestinal distress, memory impairment and even death. Coastwide closures of shellfish harvest due to this toxin have historically lasted many months, resulting in an estimated $23–28 million total loss to local fishing communities [39] and up to $11.36 million annual income loss [40], including to coastal tribes that rely on these shellfish for subsistence and income. Quinault Indian Nation tribal members depend on commercial razor clam digs for supplemental income including a special “school clothes dig” every August to provide income for households to buy clothes and supplies for the new school year. Coastwide, year-long closures that were historically common prior to establishment of the ORHAB partnership cost the Quinault Indian Nation ~$1.5 million in reduced income (Table 1), while partial year closures, made possible by ORHAB, have reduced economic loss for tribal members who rely on these shellfish sales.

By working together for the last 20 years to develop the HAB forecasting system initiated by the ORHAB partnership as its cornerstone, the PNW HAB Bulletin has allowed coastal managers to mitigate the economic effects of HABs on local economies. Tribal and non-tribal shellfish managers now use the PNW HAB Bulletin (Figure 2) to alert their HAB sampling staff to adapt baseline shore monitoring based on the Bulletin’s HAB threat status. In other words, if the Bulletin forecasts a high HAB risk prior to planned shellfish digs, managers can increase monitoring to determine which beaches are safe (or not safe) for harvests. Tribal and non-tribal agencies that manage large coastal areas can respond to these risks by allowing selective harvest of shellfish at safe beaches or increasing harvest limits prior to the arrival of forecasted HABs. This adaptive management is a direct product of the optimized monitoring program established by the ORHAB partnership, now complemented by the PNW HAB Bulletin.

## 3. Concluding Remarks

We have learned valuable lessons from developing an oceans community that includes respect for Indigenous knowledge and science which promotes a sense of unity with nature, and is not just a matter of “being close to nature” to heal ourselves. Powerfully, in the Quinault language, “ta’aWshi xa’iits’os” translates to “clam hungry”, a phrase describing the longing for and traditional dependence of the tribe on razor clams as a subsistence food. NOAA’s relationship with the Quinault Indian Nation and other coastal tribes, facilitated by the ORHAB partnership and the PNW HAB bulletin, supported the importance of preserving razor clam harvest opportunity for tribal members’ health, economic and cultural well-being. The inclusion of several coastal native communities in this partnership has provided an understanding of the role of the oceans in their lives and our collective lives and facilitated the accomplishments of the entire group of partners. The award-winning ORHAB partnership’s success and related scientific studies on coastal resilience were made possible only through the cultivation of mutual respect and understanding. (*Lesson 5, Cultivate Mutual Respect*). A reverence for nature informs our current collaborative studies on the resilience of coastal communities to climate change, using integrative social-ecological systems approaches (e.g., [42]) to address issues such as expansion of intensity and geographical scope of HABs [43], in Washington State through the SoundToxins partnership [44] and in other parts of the world, such as Alaska [45,46], Guatemala [47] and India [48]. In the end, we realize that mutual respect of humans for nature and for one another go hand in hand.

Taking to heart these lessons that non-Indigenous scientists have learned through their collaboration with Indigenous, place-based, communities, we are convinced that for global oceans and human health initiatives to be successful, we must think of ourselves as an integral part of the natural world to promote reciprocal healing [1,2,3,4,7,8]. The HAB projects described here provide a confirmation of the interdependent relationship of nature to humans and humans to nature. As Indigenous scholar and ecologist Robin Kimmerer states, “Action on behalf of life transforms. Because the relationship between self and the world is reciprocal, it is not a question of first getting enlightened or saved and then acting. As we work to heal the earth, the earth heals us” [1].

In summary, taking the long view is required for our survival and necessary to reverse the impacts of climate change and other insults on oceans, including the increasing expansion and intensity of marine debris, pathogens and HABs. Our relationship with tribal communities taught us reciprocity, highlighting both the benefits from and protection of the environment that promote our well-being. Similarly, our partnerships with one another provided opportunities for joint learning and accomplishment. Proactive action by governments, organizations and communities with close attention to social and cultural factors [49] is vital for such partnerships to be effective and impactful. Fundamentally, though, the reciprocal relationship between nature and humans begins as a partnership, stemming from meaningful relationships with other people while in nature. Whether fishing together, or collecting clams from a windy beach, each person’s attention to protective, sustained engagement with nature can create a ripple effect for conservation and build communities with a sense of solidarity, sharing, and a collective mission to care for each other and for the planet.

## Figures and Tables

**Figure 1 ijerph-18-02662-f001:**
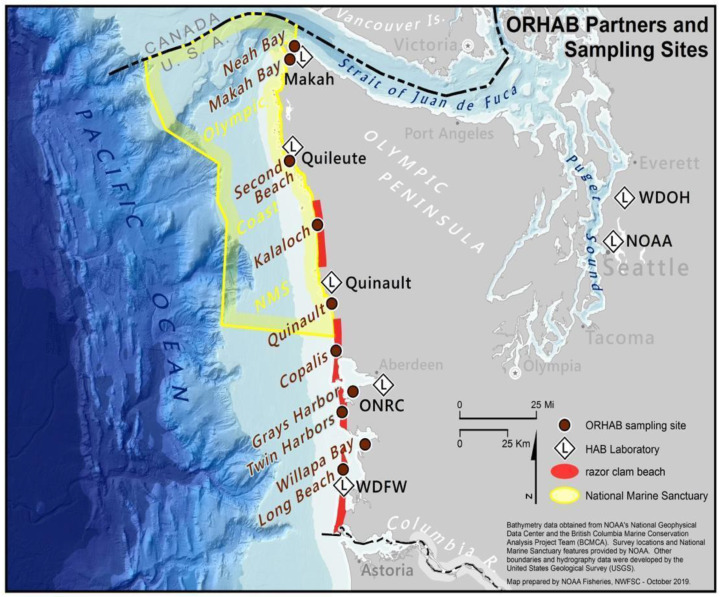
Olympic Region Harmful Algal Bloom (ORHAB) partners and sampling sites, including the Northwest Fisheries Science Center (NOAA), the Washington State Department of Health (WDOH), the Washington Department of Fish and Wildlife (WDFW), the University of Washington Olympic Natural Resources Center (ONRC), the Quinault Indian Nation (Quinault), the Makah Tribe (Makah), and Quileute Tribe (Quileute). Sampling sites, laboratories for toxin testing and phytoplankton counts, razor clam beaches, and the boundary of the National Marine Sanctuary are shown in the legend (for a full description of ORHAB, see [29].

**Figure 2 ijerph-18-02662-f002:**
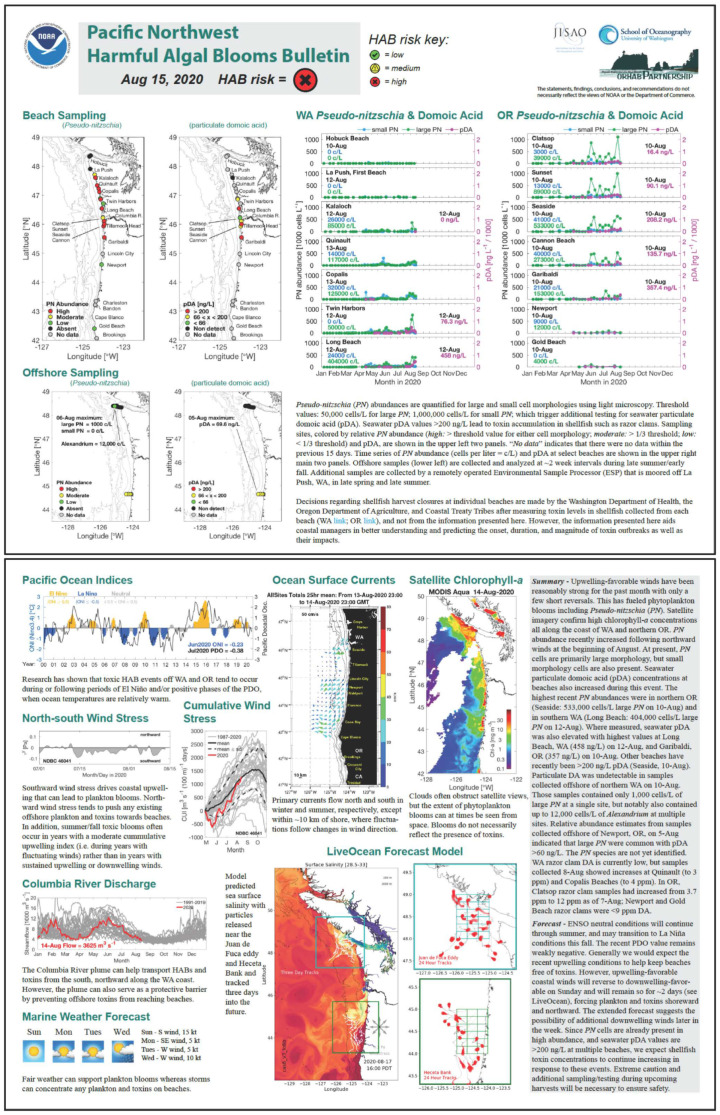
The Pacific Northwest HAB Bulletin, dated 15 August 2020, showing a high HAB risk (red X). The 2 panels of the complete Bulletin are shown here to illustrate its complexity. To view details of this Bulletin, please refer to the web version (http://www.nanoos.org/products/habs/forecasts/bulletins/pnw_hab_bulletin-20200815.pdf; accessed on 4 March 2021). This is the synthesis of several decades of research on HABs in the northeast Pacific Ocean that is responsive to the needs of coastal managers, including tribal co-managers of shellfish resources. A number of partners from universities, tribes, state management organizations and government agencies joined together from the start to write grants, conduct research, and provide input that made this HAB forecasting system a reality today. For more information about Bulletin components and design and to see more historical Bulletins, see the NANOOS website (http://www.nanoos.org/products/habs/forecasts/bulletins.php; accessed on 4 March 2021).

**Table 1 ijerph-18-02662-t001:** Economic impacts of HABs on the Quinault Indian Nation commercial fishery. Data are based on average annual harvest during 2003–2007.

Closure Type		Harvest Decline (lb)	Economic Loss (USD)
All beaches	All year	376,623	1,421,225
	Half year	188,312	710,614
Copalis beach *	All year	165,407	624,180
	Half year	82,704	312,090
Quinault beach	All year	211,216	797,045
	Half year	105,608	398,523

* locations are shown in Figure 1 (modified from [41]).

## Data Availability

Data are available in a publicly accessible repository that does not issue DOIs. Publicly available datasets were analyzed in this study. This data can be found here: www.nanoos.org; accessed on 4 March 2021.

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
