# Peer review of "Taking the Long View for Oceans and Human Health Connection through Community Driven Science"

_ijerph, 2021, doi:10.3390/ijerph18052662_

Round 1

Reviewer 1 Report

This is an excellent manuscript, written clearly, sell-organized, rich in information, and very relative to today's issues. I am pleased to recommend it for acceptance. I see only a few very minor problems:

The text citations for Box 1, Figure 1, and Figure 2 all appear after these illustrations have been presented. I suggest changing the layout, or moving the citations precede the figures. Line 300 refers to Figure 3 (HAB Bulletin). I presume the authors mean to refer to Figure 2.

Author Response

Responses in red.

This is an excellent manuscript, written clearly, sell-organized, rich in information, and very relative to today's issues. I am pleased to recommend it for acceptance. Thank you. I see only a few very minor problems:

The text citations for Box 1, Figure 1, and Figure 2 all appear after these illustrations have been presented. I suggest changing the layout, or moving the citations precede the figures. Line 300 refers to Figure 3 (HAB Bulletin). I presume the authors mean to refer to Figure 2.

We have moved the Box and Figures so that they appear after their references in the text.  The reference to Fig. 3 has been changed to “Fig. 2”.

Reviewer 2 Report

The manuscript "Taking the long view for Oceans and Human Health connection through community driven science" is an interesting presentation of an active collaboration project between the scientific community and indigenous communities to better understand environmental problems and define strategies to deal with these problems. The focus is on harmful algal blooms in the Pacific NE. It is an important approach, with points of contact with Anthropocénico and Citizen Science, two themes that have received the attention of the scientific community, which is expected to grow even further in the coming years.

With the title, one gets the idea that the work is a "Commentary", although this category of submission does not appear in the options presented by the IJERPH platform. Maybe I didn't see it well.

What is certain is that it is not a conventional article. It does not have a structure with regional framework, methods, results and discussion of results, making its assessment less linear. Assuming that this type of contribution (Commentary) is compatible with IJERPH, I just suggest an effort not to leave it so turned to self-promotion.

I left some comments in the annotated pdf, where I point out minor issues.

Author Response

Responses in red.  Please see the attachment for specific responses in the manuscript.

The manuscript "Taking the long view for Oceans and Human Health connection through community driven science" is an interesting presentation of an active collaboration project between the scientific community and indigenous communities to better understand environmental problems and define strategies to deal with these problems. The focus is on harmful algal blooms in the Pacific NE. It is an important approach, with points of contact with Anthropocénico and Citizen Science, two themes that have received the attention of the scientific community, which is expected to grow even further in the coming years.

With the title, one gets the idea that the work is a "Commentary", although this category of submission does not appear in the options presented by the IJERPH platform. Maybe I didn't see it well.

Thank you for your comment.  In fact, Commentary is a submission option for IJERPH.  See for example these Commentaries: https://www.mdpi.com/search?q=commentary&journal=ijerph

What is certain is that it is not a conventional article. It does not have a structure with regional framework, methods, results and discussion of results, making its assessment less linear. Assuming that this type of contribution (Commentary) is compatible with IJERPH, I just suggest an effort not to leave it so turned to self-promotion.

I left some comments in the annotated pdf, where I point out minor issues.

We have responded to the Reviewer’s comments directly on the pdf.  Thank you for your helpful comments.

Reviewer 3 Report

I personally liked this commentary very much. I think this is an essential approach for coastal management. I congratulate this effort and hope this information serves as a model for other communities around the world.

Showing respect for ancient knowledge and allowing our scientific egos to learn from native people is imperative, as is to share what we know and work together. Lesson 2, I think, is crucial and often overlooked.

The development and support of internal monitoring programs working cooperatively with state and academic partners is a dream come true! The biocultural approach is urgent, and I applause this initiative!

The flawed idea of “being close to nature to heal ourselves”, which is selfish and ineffective, should be changed for the concept of learning to live all together in this earth as connected factors that could heal or make ill each other. So, congratulations for this work.

Nevertheless, some details should be revised:

Line 46: check spelling.

Line 81: the authors use the term western for non-Indigenous, but then by the end of the manuscript they call “non-Indigenous”. I think this term, “western”, should be avoided because there are also western indigenous communities, maybe not in the geographical place where this particular project was developed, but in other areas. The term “non-indigenous” is adequate.

Line 180 – on: it is not clear if the people who depend on shellfish are or are not more well-adapted to shellfish poisoning. I think it is necessary to clarify this issue.

Figure 2: these are 2 different figures, they are too small, impossible to read and the image quality is not enough to make it bigger on screen. Also, the legend -which is inside a box, is not clear.

Line 261: I think there is a comma missing before lives, but I suggest clarifying this sentence.

Table 1: check and improve formatting.

Line 300: there is no Fig. 3 in the paper.

Line 321: the superscript is written twice.

There are several formatting details that have to be taken care of.

About the “author credentials”, I wonder, more than the credentials, the author’s contributions to the paper.

Author Response

Responses in red.

I personally liked this commentary very much. I think this is an essential approach for coastal management. I congratulate this effort and hope this information serves as a model for other communities around the world.

Thank you very much for your kind words.

Showing respect for ancient knowledge and allowing our scientific egos to learn from native people is imperative, as is to share what we know and work together. Lesson 2, I think, is crucial and often overlooked.

The development and support of internal monitoring programs working cooperatively with state and academic partners is a dream come true! The biocultural approach is urgent, and I applause this initiative!

The flawed idea of “being close to nature to heal ourselves”, which is selfish and ineffective, should be changed for the concept of learning to live all together in this earth as connected factors that could heal or make ill each other. So, congratulations for this work.

Nevertheless, some details should be revised:

Line 46: check spelling.

Thank you.

Line 81: the authors use the term western for non-Indigenous, but then by the end of the manuscript they call “non-Indigenous”. I think this term, “western”, should be avoided because there are also western indigenous communities, maybe not in the geographical place where this particular project was developed, but in other areas. The term “non-indigenous” is adequate.

We have changed the words “western scientists” to “non-Indigenous scientists” throughout the manuscript.

Line 180 – on: it is not clear if the people who depend on shellfish are or are not more well-adapted to shellfish poisoning. I think it is necessary to clarify this issue.

We have clarified with this sentence: These stories suggest that the visitors became ill from consuming shellfish which contained toxins, while the Makah, who habitually rely on shellfish in their diet, did not. AND These traditional stories provided inspiration to further our understanding of genetic resistance to toxins in shellfish that consume toxic algae.

Figure 2: these are 2 different figures, they are too small, impossible to read and the image quality is not enough to make it bigger on screen. Also, the legend -which is inside a box, is not clear.

We have added the sentence “The 2 panels of the complete Bulletin are shown here to illustrate its complexity.  To view details of this Bulletin, please refer to the web version.”

Line 261: I think there is a comma missing before lives, but I suggest clarifying this sentence.

We have modified this to read: “In the specific case of HABs, it is critical to identify the sites that serve as the source of these blooms while learning how environmental factors (e.g., wind, temperature, nutrient loading, and pH) affect the drivers of ocean challenges. This example demonstrates a paradigm shift from crisis management to a longer, better informed, ‘up-stream’ response which in the end will save resources, and protect the overall health of humans and oceans.”

Table 1: check and improve formatting.

The Table format has been improved

Line 300: there is no Fig. 3 in the paper.

Thank you.  This has been changed to “Fig. 2”.

Line 321: the superscript is written twice.

Thank you.

There are several formatting details that have to be taken care of.

About the “author credentials”, I wonder, more than the credentials, the author’s contributions to the paper.

We have changed this to read “Author Contributions” and have removed the section about author credentials.

Round 2

Reviewer 2 Report

I acknowledge that the authors took my suggestions into account when reviewing the article. I can only hope that these initiatives involving indigenous communities will succeed and benefit everyone.